# Effect of Adenomatous Polyposis Coli Loss on Tumorigenic Potential in Pancreatic Ductal Adenocarcinoma

**DOI:** 10.3390/cells8091084

**Published:** 2019-09-14

**Authors:** Jennifer M. Cole, Kaitlyn Simmons, Jenifer R. Prosperi

**Affiliations:** 1.Department of Biochemistry and Molecular Biology, Indiana University School of Medicine, South Bend, IN 46617, USA; jennifermichelecole@gmail.com; 2.Mike and Josie Harper Cancer Research Institute, South Bend, IN 46617, USA; ksimmon1@alumni.nd.edu; 3.Department of Biological Sciences at University of Notre Dame, South Bend, IN 46617, USA

**Keywords:** PDAC, APC, proliferation, migration, gemcitabine

## Abstract

Loss of the *Adenomatous Polyposis Coli* (*APC*) tumor suppressor in colorectal cancer elicits rapid signaling through the Wnt/β-catenin signaling pathway. In contrast to this well-established role of APC, recent studies from our laboratory demonstrated that APC functions through Wnt-independent pathways to mediate in vitro and in vivo models of breast tumorigenesis. Pancreatic ductal adenocarcinoma (PDAC) has an overall median survival of less than one year with a 5-year survival rate of 7.2%. APC is lost in a subset of pancreatic cancers, but the impact on Wnt signaling or tumor development is unclear. Given the lack of effective treatment strategies for pancreatic cancer, it is important to understand the functional implications of APC loss in pancreatic cancer cell lines. Therefore, the goal of this project is to study how *APC* loss affects Wnt pathway activation and in vitro tumor phenotypes. Using lentiviral shRNA, we successfully knocked down APC expression in six pancreatic cancer cell lines (AsPC-1, BxPC3, L3.6pl, HPAF-II, Hs 766T, MIA PaCa-2). No changes were observed in localization of β-catenin or reporter assays to assess β-catenin/TCF interaction. Despite this lack of Wnt/β-catenin pathway activation, the majority of *APC* knockdown cell lines exhibit an increase in cell proliferation. Cell migration assays showed that the BxPC-3 and L3.6pl cells were impacted by APC knockdown, showing faster wound healing in scratch wound assays. Interestingly, APC knockdown had no effect on gemcitabine treatment, which is the standard care for pancreatic cancer. It is important to understand the functional implications of APC loss in pancreatic cancer cells lines, which could be used as a target for therapeutics.

## 1. Introduction

Pancreatic ductal adenocarcinoma (PDAC) has an overall median survival of less than one year and is the third leading cause of cancer-related deaths for both men and women in the United States [1]. Patients are often asymptomatic early on, which results in approximately 53% of patients being diagnosed in advanced stages. The five-year survival rate for advanced stage PDAC is 7.2% [2]. Although researchers are actively investigating early detection methods, there are currently no reliable methods for detecting pancreatic cancer (Cancer Facts and Figures 2015). In contrast to the increased survival of patients with many other cancers, advancements in pancreatic cancer treatment have been slow [1]. In fact, death rates due to pancreatic cancer are rising for both men and women [1]. Treatments for patients with PDAC include surgery or chemotherapy treatments using a DNA nucleoside, gemcitabine [3]. Although these treatment options may relieve symptoms or extend survival, they seldom produce a cure. Because pancreatic cancer is typically detected after metastasis has occurred, less than 20% of patients are candidates for surgery.

The APC tumor suppressor is a potent negative regulator of the Wnt/β-catenin signaling pathway. Mutations in *APC* were first studied in colorectal cancer [4,5,6,7]. APC inactivation has been found in approximately 35% to 88% of colorectal tumors, making it the most common genetic alteration observed in colorectal cancers [8]. Recent studies have also identified APC mutations in many epithelial cancers, including breast and lung cancer (reviewed in [9]). In some extracolonic tumors, including pancreatic [10,11], inactivation of APC occurs through promoter methylation and/or results in Wnt-independent signaling mechanisms [9], suggesting a tissue-specific effect of APC on tumor development. The importance of APC in pancreatic cancer is not yet fully understood, and appears complicated depending on the type of pancreatic cancer being assessed [11,12]. APC was methylated in 58.6% of PDAC, with prevalence of APC methylation increasing with tumor progression [13]. In another study, somatic mutations in *APC* were observed in 4 of 10 pancreatic tumors examined [14]. Of these, two tumors contained mutations in the mutation cluster region (MCR), which includes the β-catenin binding domain. These frameshift mutations were caused by single base pair deletions, leading to a truncated protein and loss of function [14]. Familial adenomatous polyposis (FAP) is caused by a mutation in the tumor suppressor, APC, and has been linked to patients with pancreatic cancer [15,16,17]. One study collected data from The Johns Hopkins Polyposis Registry, and found 4/1391 patients with FAP who developed extraintestinal cancer in the pancreas, with a relative risk (observed/expected) of 4.5 when compared to the general population [15]. Patients with FAP have demonstrated intraductal papillary and mucinous pancreatic tumors, and high-grade pancreatic intraepithelial neoplasia, a precursor to invasive ductal carcinoma [17,18]. 

Given that not much is known about APC in PDAC, the impact of APC loss on Wnt/β-catenin signaling and tumor development in PDAC is unclear. It is important to understand the functional implications of APC loss in pancreatic cancer cells lines. Our research investigates whether APC loss in pancreatic cancer mediates in vitro tumorigenic potential. The studies herein describe the effect of APC loss on PDAC cell proliferation, migration, and response to gemcitabine. 

## 2. Materials and Methods

### 2.1. Cells and Lentiviral Transductions

Six pancreatic cancer cell lines (MIA PaCa-2, BxPC-3, L3.6pl, Hs 766T, AsPC-1, and HPAF-II) were received from Dr. Reginald Hill (previously at University of Notre Dame; now at USC), and were used for these studies. MIA PaCa-2, BxPC-3 and L3.6pl pancreatic cancer cell lines, and control SW480 and MCF-7 cells were maintained in DMEM supplemented with 10% fetal bovine serum, 1% penicillin/streptomycin and 5 μg/mL plasmocin (InvivoGen, San Diego, CA, USA). Hs 766T, AsPC-1, and HPAF-II cells were maintained in RPMI 1640 media supplemented with 10% fetal bovine serum, 1% penicillin/streptomycin and 5 μg/mL plasmocin. The BxPC-3 cells have been previously shown to have moderate APC expression [19]. While APC expression has not been investigated in all cell lines, a previous investigation showed a lack of Wnt pathway activation in the AsPC-1, BxPC-3, Hs 766T, and MIA-PaCa-2 cells, suggesting intact APC expression [20]. All cells were routinely passaged using 0.25% trypsin/EDTA and maintained at 37 °C with 5% CO_2_. Lentiviral mediated shRNA knockdown of *APC* was obtained using two different MISSION shRNA *APC* constructs (Sigma-Aldrich, St Louis, MO, USA), with pLKO.1 or the SHC002 scrambled vector (Sigma-Aldrich) as the control. *APC* knockdown was maintained in each cell line using puromycin (1 μg/mL for BxPC-3, L3.6pl, HPAF-II, and AsPC-1, 0.5 μg/mL for MIA PaCa-2, and 3 μg/mL for Hs 766T) (Sigma-Aldrich).

### 2.2. Real-Time PCR

RNA was isolated using TriReagent (Molecular Research Center, Cincinnati, OH, USA). cDNA synthesis was performed with iScript from 1 μg RNA (BioRad Laboratories, Hercules, CA, USA). The knockdown of *APC* was quantified using RT-PCR using Power SYBR Green Master Mix (Applied Biosystems, Foster City, CA, USA), 1 μg of cDNA, and 7.5 μM of each primer (5′ to 3′ forward primer of *APC* TGTCCCGTTCTTATGGAA and 5′ to 3′ reverse primer of *APC* TCTTGGAAATGAACCCATAGG) and CFX Connect 96 thermal cycler (Bio-Rad Laboratories). Cycling conditions were 50 °C for 2 min, 95 °C for 10 min, 40 cycles of 95 °C for 15 s, and 60 °C for 1 min. Glyceraldehyde 3-phosphate dehydrogenase (*GAPDH*) was used as a reference gene for these experiments. Samples were analyzed in duplicate and values were averaged. 

### 2.3. Immunofluorescence

Cells were plated on coverslips in duplicate at a density of 1 × 10^5^ cells per well in 6-well plates and grown to 50%–70% confluence. Cells were then fixed with 3.7% paraformaldehyde and permeabilized with 0.3% Triton X-100. Staining was performed with β-catenin mouse primary antibody (BD Laboratories, San Diego, CA, USA) diluted 1:400 in blocking buffer. Expression was detected using Alexa-conjugated β-catenin goat, anti-mouse 488 secondary antibodies (1:1000 each; Life Technologies, Carlsbad, CA, USA). For visualization of F-actin, cells were co-stained with Alexa-conjugated Phalloidin 594 (1:200; Life Technologies). All slides were mounted with Fluoromount G (SouthernBiotech, Birmingham, AL) with Hoechst and 63× oil images were obtained.

### 2.4. β-Catenin/TCF Reporter Assays 

Cells were plated in triplicate at a density of 1 × 10^5^ cells per well in 24-well plates and transfected with pTOPflash or pFOPflash, and co-transfected with pRL-TK for transfection efficiency control (Promega, Madison, WI, USA). SW480, MCF7, and L3.6pl cells were transfected with Lipofectamine 2000 (Invitrogen, Carlsbad, CA, USA) as per the manufacturer’s instructions. MIA PaCa-2, BxPC-3, and AsPC-1 cells were transfected using Trans-IT 2020 (Mirus Bio, Madison, WI, USA) as per the manufacturer’s instructions. Hs 766T cells were transfected with Optifect (Thermo Fisher Scientific, Rockford, IL) and HPAF-II cells were transfected with FuGENE HD (Promega) as per the manufacturer’s instructions. Lysates were harvested after 48 h and analyzed using the Dual Luciferase Assay System kit (Promega). Luciferase activity was normalized for transfection efficiency and FOPflash activity as previously described [9].

### 2.5. Cell Growth Assay

Cells were plated in triplicate at a density of 5 × 10^4^ cells/well in 12-well plates and counted at 24, 48, and 72 h.

### 2.6. Cell Migration Assay

Cells were plated at a density of 1 × 10^5^ cells/well in a 6-well plate and grown to confluence. Upon reaching confluence, cells were treated with 2.5 μg/mL mitomycin C (Sigma, St Louis, MI, USA) to inhibit proliferation, and three scratch marks were made per well with a 0.1–10 μL pipet tip. Images were taken at 0, 12, 24, and 48 h post-wounding and analyzed with T-Scratch software version 1.0 (CSElab at ETH Zurich, [21]).

### 2.7. Gemcitabine Assay

Cells were plated in duplicate per condition at a density of 1 × 10^5^ cells/well in 6-well plates and grown to 50% confluence. Cells were then treated with 1 μM gemcitabine (Sun Pharmaceutical Ind. Ltd., Gujarat, India) or vehicle control and counted 48 h post-treatment.

### 2.8. Statistics

All experiments were performed three independent times, and analyzed for significance using a standard *t*-test. All data with a *p*-value of less than 0.05 are considered statistically significant.

## 3. Results and Discussion 

### 3.1. Verification of APC Knockdown in Pancreatic Ductal Adenocarcinoma Cell Lines

Using lentiviral-mediated MISSION shRNA (Sigma), we knocked down *APC* in six pancreatic ductal adenocarcinoma cell lines (AsPC-1, BxPC-3, HPAF-II, Hs 766T, L3.6pl, and MIA PaCa-2). AsPC-1 pancreatic cancer cells were derived from nude mouse xenografts initiated with cells from ascites of a patient with cancer of the pancreas. The BxPC-3 adenocarcinoma cells were derived from a primary pancreatic tumor. HPAF-II are human adenocarcinoma cells derived from peritoneal ascites fluid from a male with primary pancreas adenocarcinoma with metastasis to the liver, diaphragm and lymph nodes. Hs 766T are pancreatic carcinoma cells derived from lymph nodes. L3.6pl are a metastatic pancreatic cancer line. MIA PaCa-2 are carcinoma cells derived from tumor tissue of the pancreas. We chose these cell lines because they provided the primary and metastatic backgrounds of PDAC and have varying response to gemcitabine. To verify knockdown, we used quantitative RT-PCR to determine the relative expression of *APC*, which revealed knockdown in most of the cell lines compared to their respective controls (Figure 1A–F). 

### 3.2. APC Knockdown in PDAC Cells Does Not Elicit Signaling through Wnt/Β-Catenin

In the APC knockdown PDAC cell lines, we performed immunofluorescence (IF) for β-catenin and showed no changes in localization after APC knockdown (Figure 2A). In addition, we observed no alterations in Wnt/β-catenin signaling using β-catenin/TCF reporter assays (Figure 2B). In both assays, the SW480 human colorectal cancer cells were used as a positive control given their enhanced Wnt/β-catenin signaling. MCF-7 human breast cancer cells were used as a negative control for the reporter assay, as they show minimal Wnt signaling. We have also investigated common Wnt target genes (BIRC5, c-myc, and cyclin D1), and showed no changes after APC knockdown (data not shown). These results, showing lack of activation of the Wnt/β-catenin signaling pathway, are similar to our studies in breast cancer cells and MDCK cells with APC knockdown [9,22,23]. While inhibitors of Wnt/β-catenin have shown promise in in vitro studies [24], our results suggest that APC may have Wnt-independent functions in pancreatic cancer regulation. 

### 3.3. APC Loss Enhances Proliferation in PDAC Cells

APC loss has been shown to enhance proliferation in breast and colon cancer, amongst others [9,25,26]. Therefore, we assessed whether proliferation was altered in APC knockdown PDAC cells. Cell growth was measured via total cell counts at 24, 48 and 72 h post-plating for each cell line. We demonstrated that APC knockdown significantly increased proliferation in BxPC-3, Hs 766T and L3.6pl cell lines compared to their individual controls (Figure 3B,D,E). The AsPC1, HPAF-II and MIA PaCa-2 cell lines showed no significant differences in cell proliferation of the APC shRNA cells lines compared to the control (Figure 3A,C,F). These three cell lines could be proliferating quickly already and therefore see no differences in APC knockdown compared to the control. 

Our data show similar doubling time as previously published [27]. AsPC-1 is a pancreatic adenocarcinoma cell line derived from a metastatic site (ascites) with a doubling time of 38–40 h [28]. BxPC-3 are primary pancreatic adenocarcinomas with a doubling time of 48–60 h [29]. HPAF-II is a human pancreatic adenocarcinoma cell line derived from a metastatic site (ascites) with a doubling time of 42 h [30]. Hs 766T are pancreatic carcinoma cells derived from a metastatic site (lymph node) with a doubling time of 6–7 days [31]. L3.6pl is a pancreas cell line derived from a liver metastasis. Cell division genes have been shown to be upregulated significantly in gene expression profiles generated from microarrays of the highly metastatic cell line L3.6pl [32]. MIA Paca-2 is a primary tumor with a doubling time of 40 h [33]. 

### 3.4. Effect of APC Knockdown on PDAC Cell Migration 

Given that tumor metastasis is associated with a poor prognosis, we next sought to investigate cell migration using a wound-healing assay. Cells were grown to confluence, scratched, imaged, and analyzed at 0, 24, and 48 h. Using the T-scratch software, we determined the area filled in to represent cell migration [21]. No significant differences were observed in HPAF-II, Hs766T, and MIA PaCa-2 cells, in percent filled area of APC knockdown cells versus the control (Figure 4C,D,F). While the AsPC-1 APC shRNA1 cells showed no changes, the APC shRNA2 cells migrated slower than the CTL cells (Figure 4A). APC knockdown in the BxPC-3 and L3.6pl cells resulted in faster wound healing (Figure 4B,E), suggesting that these cells become more aggressive with APC knockdown. Specifically, 48 h post-wounding of the BxPC-3 cells, the APC shRNA1 and APC shRNA2 cell lines were significantly more closed than the CTL cells (Figure 4B). In L3.6pl, there were significant differences in percent filled area in APC shRNA1 and APC shRNA2 at 24 h and APC shRNA1 at 48 h compared to CTL cells (Figure 4E). In our studies, we found that the BxPC-3 cells with APC knockdown had filled the wound area more at 24 and 48 h compared to the control. BxPC-3 cells are a primary cell line derived from a patient that had no signs of metastasis. Very few studies have looked at the migration patterns of multiple PDAC cell lines; however, one study showed BxPC-3 had a relative wound density of around 80% at 48 h and reached complete closure at 60 h post-wounding [34].

### 3.5. APC Knockdown Does Not Impact PDAC Cell Response to Gemcitabine 

Treatments for patients with PDAC include chemotherapy using the DNA nucleoside, gemcitabine [3]. We previously demonstrated that APC loss results in resistance to chemotherapeutic agents in mouse and human breast cancer cells [25]. Therefore, we sought to investigate the effect of APC knockdown in pancreatic cancer cells on response to gemcitabine. Cells were treated with 1 µM gemcitabine, and overall cell number was assessed at 24 (data not shown) and 48 h. While gemcitabine decreased cell number in most cell lines, there was no impact of APC knockdown on response to treatment. In AsPC-1 cells, we saw significant decreases in total cell numbers in controls and APC shRNA1 when treated with 1 µM gemcitabine for 48 h (Figure 5A). In BxPC-3 cells, 1 µM gemcitabine decreased total cell numbers in all cell lines (Figure 5B). In HPAF-II cells, 1 µM gemcitabine decreased total cell numbers in pLKO.1, APC shRNA1, and APC shRNA2, but not in CTL cells (Figure 5C). In Hs 766T cells, 48 h treatment with 1 µM gemcitabine resulted in significant decreases in total cell numbers in controls and APC shRNA1 (Figure 5D). In L3.6pl and MIA PaCa-2 cells, 1 µM gemcitabine decreased total cell numbers in all cell lines (Figure 5E,F). Our studies showed that all cell line controls and their APC knockdown cell lines were sensitive to 1 µM gemcitabine except for the HPAF-II control and two APC shRNA lines in the AsPC-1 and Hs 766T cell lines. L3.6pl and BxPC-3 are sensitive to, and Hs 766T, AsPC-1 and MIA PaCa-2 have been shown to be resistant to, 10 µmol/L gemcitabine [35]. This suggests that APC status would not serve as a marker for response to gemcitabine. A range of gemcitabine doses has been tested on multiple cell lines, including BxPC-3 and MIA-Paca-2, and it was observed that MIA PaCA-2 cells were more sensitive than BxPC-3 with an IC_50_ of 61 ± 3 mM versus 128 ± 16 mM, respectively [36]. 

## 4. Conclusions

In the current study, we successfully knocked down APC in six pancreatic ductal adenocarcinoma cell lines; AsPC-1, BxPC-3, HPAF-II, Hs 766T, L3.6pl, and MIA PaCa-2. β-catenin localization was at the cell–cell junctions and not nuclear, suggesting that the Wnt pathway is not active, which is similar to previously published data [20]. This was confirmed by reporter assays to assess β-catenin/TCF interaction. We saw only one of the AsPC-1 knockdown lines that was elevated in the TCF reporter assay. Proliferation was significantly increased after APC knockdown in BxPC-3 and L3.6pl at 48 h and in Hs 766T at 48 and 72 h compared to CTL. Wound healing was decreased after APC knockdown in one of the AsPC-1 knockdown lines at 24 and 48 h. In the BxPC-3 and L3.6pl knockdown lines, we saw faster wound healing, as expected with APC knockdown. Responses to gemcitabine, a standard pancreatic cancer chemotherapeutic agent, showed sensitivity in most CTL and APC shRNA lines, but was not dependent on APC status. These results are summarized in Table 1. Of the three cell lines showing phenotypic changes with APC knockdown (BxPC-3, Hs 766T, L3.6pl), there are no overt similarities in known oncogenic mutations. A search of the COSMIC database and literature showed no pattern of mutations in KRAS, SMAD4, CDKN2A, or p53. The Hs 766T and L3.6pl cells both have mutant KRAS and WT p53, while the Bx-PC3 cells have WT KRAS and mutant p53. Of the three cell lines, only the Hs 766T cells have mutant CDKN2A. This suggests that further investigation into the cell lines is required to uncover the mechanism by which APC status controls proliferation and migration in pancreatic cancer cells. Previous studies have shown a link between p53 or BRCA status and APC in pancreatic cancer [37,38], suggesting that APC works in concert with other oncogenic mutations to drive pancreatic tumorigenesis. Given the variable results observed in different cell lines, future studies would be necessary to understand the crosstalk between APC loss and other drivers of pancreatic cancer. In addition, the possibility of using APC status as either a prediction marker for pancreatic cancer or a tool to determine response to therapy should be explored [39,40]. 

## Figures and Tables

**Figure 1 cells-08-01084-f001:**
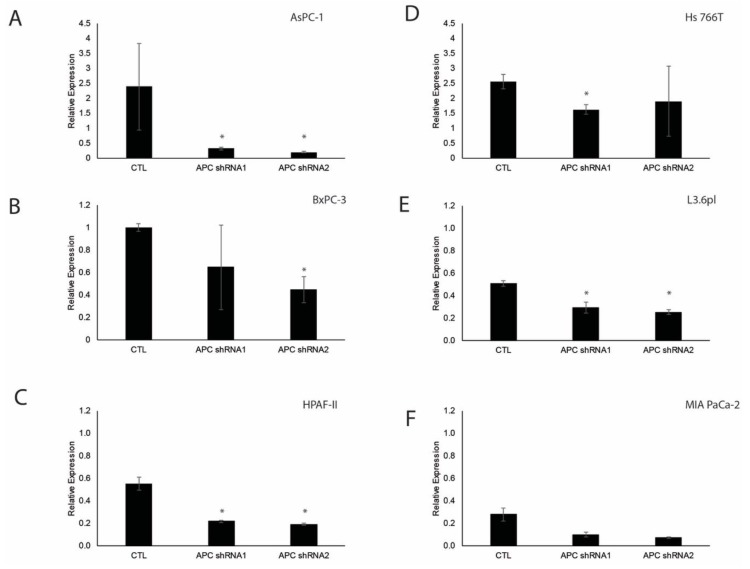
Quantitative PCR demonstrates that APC is knocked down in the cell lines transfected with APC shRNA constructs. Using lentiviral-mediated MISSION shRNA we verified *APC* knock down in six pancreatic ductal adenocarcinoma cell lines using quantitative PCR and relative expression of *APC* revealed gene knockdown in AsPC-1, BxPC-3, HPAF-II, Hs 766T, L3.6pl, and MIA PaCa-2 cell lines compared to CTL (Figure 1**A**–**F**). **p* < 0.05.

**Figure 2 cells-08-01084-f002:**
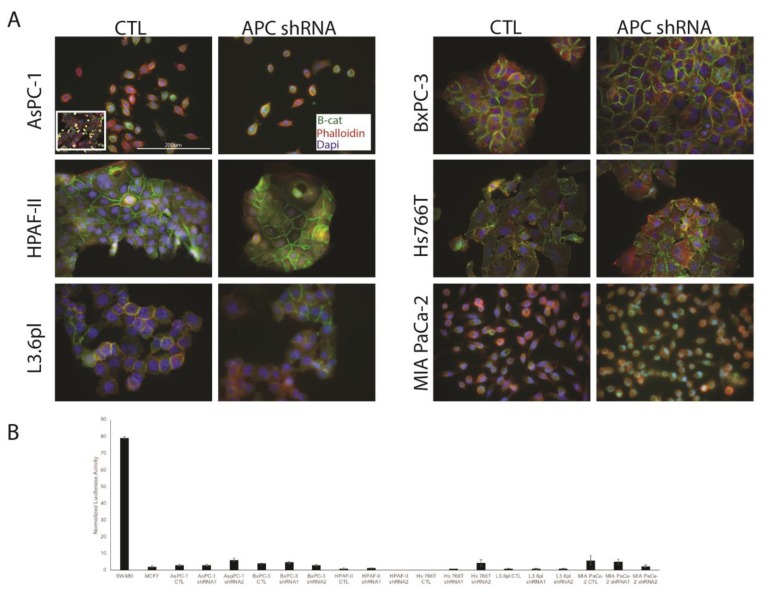
APC knockdown did not increase nuclear β-catenin. (**A**) In AsPC-1, BxPC-3, HPAF-II, Hs 766T, L3.6pl and MIA PaCa-2 CTL cells, β-catenin is localized at the cell–cell junctions. After the knockdown of APC in each of those cell lines: APC shRNA (representative), there were no changes in beta catenin localization. (**B**) In the APC knockdown PDAC cell lines, we observed no alterations in Wnt/β-catenin signaling using β-catenin/TCF reporter assays.

**Figure 3 cells-08-01084-f003:**
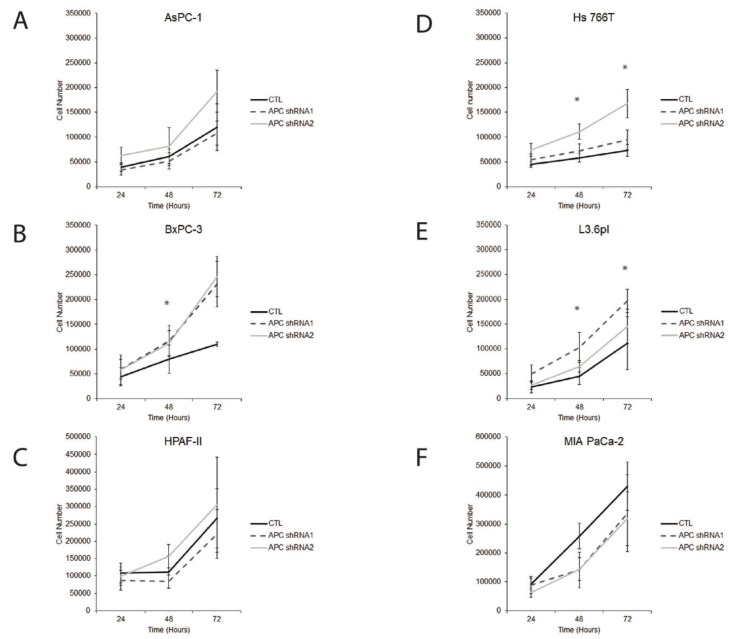
Proliferation changes are observed in some cell lines that have been transfected with APC shRNA constructs compared to controls. Proliferation was measured in total cell counts at 24, 48 and 72 h post-plating for each cell line. (**B** and **E**) In the BxPC-3 and L3.6pl cells, knockdown of APC resulted in a significant increase in cell proliferation at 48 h. (**D**) In the Hs 766T cells, knockdown of APC resulted in a significant increase in cell proliferation at 48 and 72 h. (**A**, **C**, and **F**) The AsPC-1, HPAF-II and MIA PaCa-2 cell lines showed no significant differences in cell proliferation of the APC shRNA cells lines compared to CTL. **p* < 0.05.

**Figure 4 cells-08-01084-f004:**
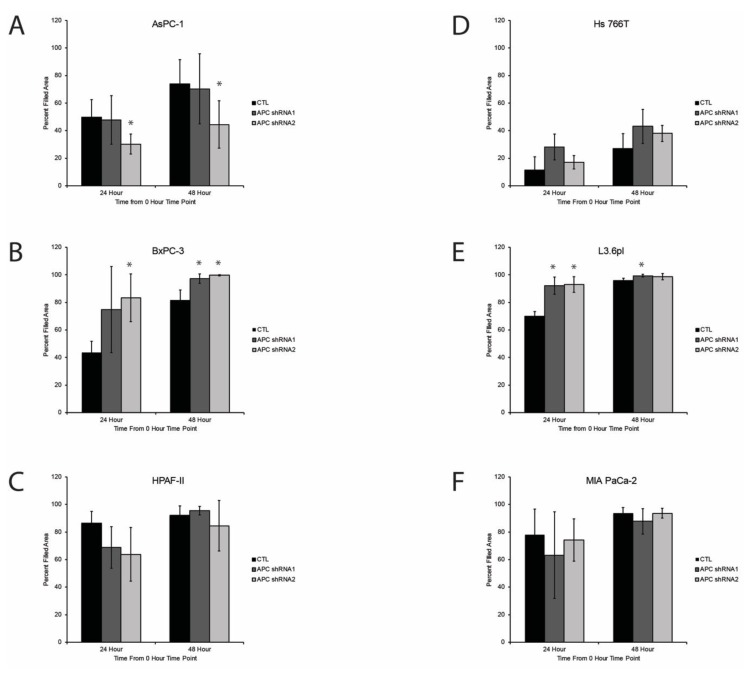
Migration of cells in wound healing assays revealed faster migration with APC knockdown in some pancreatic cancer cell lines. (**A**) The AsPC-1 APC shRNA2 knockdown cells migrated slower than the control at 24 and 48 h post wounding. (**B** and **E**) Only the BxPC-3 and L3.6pl cells were impacted by APC knockdown, showing faster wound healing. Specifically, at 48 h post-wounding, the APC shRNA1 and APC shRNA2 cell lines were significantly more closed than the parent BxPC-3 cells. (**E**) In L3.6pl, there were significant differences in percent filled area in APC shRNA1 and APC shRNA2 at 24 h and APC shRNA2 at 48 h compared to CTL. (**A**, **C**, **D**, and **F**) In AsPC-1, HPAF-II, Hs766T, and MIA PaCa-2 cells, there were no significant differences in percent filled area in APC shRNA cells versus CTL. **p* < 0.05.

**Figure 5 cells-08-01084-f005:**
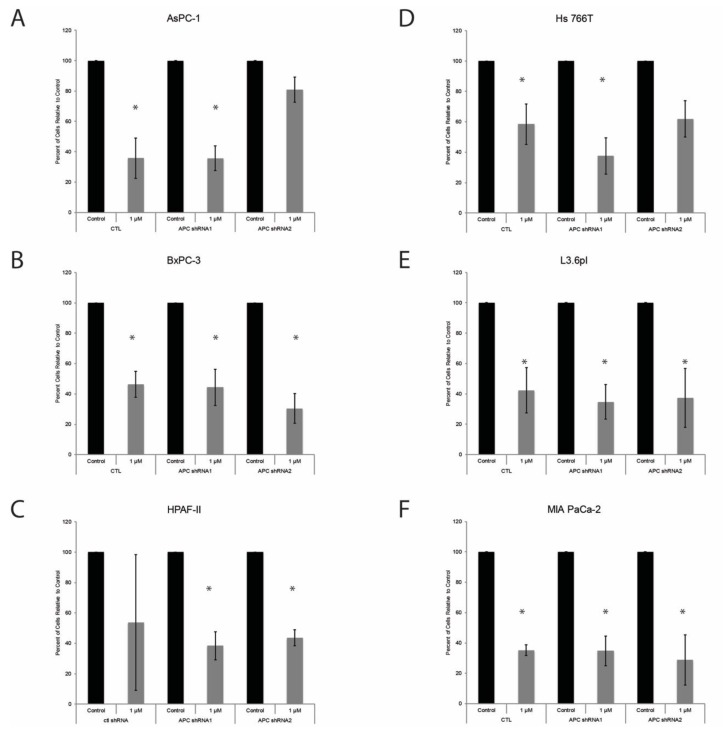
Effect of gemcitabine on PDAC cell lines. (**A**) In AsPC-1 cells, we saw significant decreases in total cell numbers in controls and APC shRNA1, but not in APC shRNA2 when treated with 1 µM gemcitabine for 48 h. (**B**) In BxPC-3 cells, 1 µM gemcitabine decreased total cell numbers in all cell lines. (**C**) In HPAF-II cells, 1 µM gemcitabine decreased total cell numbers in APC shRNA1, and APC shRNA2, but not in the CTL. (**D**) In Hs 766T cells, 48 h treatment with 1 µM gemcitabine resulted in significant decreases in total cell numbers in controls and APC shRNA1, but not in APC shRNA2. (**E**) In L3.6pl cells, 1 µM gemcitabine decreased total cell numbers in CTL APC shRNA1, and APC shRNA2. (**F**) In MIA PaCa-2 cells, 1 µM gemcitabine decreased total cell numbers in CTL, APC shRNA1, and APC shRNA2. **p* < 0.05.

**Table 1 cells-08-01084-t001:** Summary of data.

	β-Catenin Localization	Proliferation	Wound Healing	Response to Gemcitabine
**AsPC-1 CTL**	Junctional	Doubling time of approximately 24–48 h.	Wound closed approximately 73% at 48 h.	Significantly decreased total cell number compared to untreated.
APC shRNA1	NC	NC	NC	NC
APC shRNA2	NC	NC	Wound closed significantly slower compared to CTL at 24 and 48 h.	No significance differences compared to untreated.
**BxPC-3 CTL**	Junctional	Doubling time of approximately 24 h.	Wound closed approximately 81% at 48 h.	Significantly decreased total cell number compared to untreated.
APC shRNA1	NC	NC	Wound closed significantly faster compared to CTL at 48 h.	NC
APC shRNA2	NC	Significantly different from CTL at 48 h.	Wound closed significantly faster compared to CTL at 24 and 48 h.	NC
**HPAF-II CTL**	Junctional	Doubling time of approximately 48 h.	Wound closed approximately 92% at 48 h.	No significance differences compared to untreated.
APC shRNA1	NC	NC	NC	Significantly decreased total cell number compared to untreated.
APC shRNA2	NC	NC	NC	Significantly decreased total cell number compared to untreated.
**Hs 766T CTL**	Junctional	Doubling time greater than our 72 h experimental timeframe.	Wound closed approximately 27% at 48 h.	Significantly decreased total cell number compared to CTL.
APC shRNA1	NC	NC	NC	NC
APC shRNA2	NC	Significantly different from CTL at 48 and 72 h.	NC	No significance differences compared to untreated.
**L3.6pl CTL**	Junctional	Doubling time of approximately 24 h.	Wound closed approximately 95% at 48 h.	Significantly decreased total cell number compared to untreated.
APC shRNA1	NC	Significantly different from CTL at 48 h.	Wound closed significantly faster compared to CTL at 24 and 48 h.	NC
APC shRNA2	NC	NC	Wound closed significantly faster compared to CTL at 24 h.	NC
**MIA-PaCa CTL**	Junctional	Doubling time of approximately 30 h.	Wound closed approximately 93% at 48 h.	Significantly decreased total cell number compared to untreated.
APC shRNA1	NC	NC	NC	NC
APC shRNA2	NC	NC	NC	NC

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
