# Peer review of "Effect of Adenomatous Polyposis Coli Loss on Tumorigenic Potential in Pancreatic Ductal Adenocarcinoma"

_cells, 2019, doi:10.3390/cells8091084_

Round 1

Reviewer 1 Report

Thank you for providing modified version of manuscript. Revised manuscript has been considerably improved following the referee's suggestions, which have been precisely addressed by the authors when providing a major revision of the manuscript. Authors prepared a careful revision of their manuscript. Technically the work is well done. No weak data were found.

Author Response

Thank you for the review and recognizing the improvements that have been made. 

Reviewer 2 Report

In the present manuscript the authors have explored the functional implications of APC loss in pancreatic cancer cells lines. After APC knockdown using lentiviral mediated MISSION shRNA, they performed several in vitro functional assays to evaluate the tumorigenic potential of APC loss in this cancer type. In this new version they have revised and improved their manuscript before the resubmission. However, there are some additional points that need to be addressed. Specifically, the authors should discuss their findings in the examined cell lines according to their mutational status for APC and other key genes in pancreatic cancer progression (such as KRAS, SMAD4, CDKN2A and TP53). Currently, these data are freely available on several online websites (see for example COSMIC database). Conceivably, this information could also explain the observed independence from the canonical Wnt pathway (see for example Wang et al., 2015, Cell 163, 1237-1251). Beside, in the methods, the source of the utilized cell lines should be stated.

Author Response

"The authors should discuss their findings in the examined cell lines according to their mutational status for APC and other key genes in pancreatic cancer progression (such as KRAS, SMAD4, CDKN2A and TP53). Currently, these data are freely available on several online websites (see for example COSMIC database). Conceivably, this information could also explain the observed independence from the canonical Wnt pathway (see for example Wang et al., 2015, Cell 163, 1237-1251)." We apologize for the oversight of including this information. We had initially looked at the KRAS mutation status. Since we didn't notice any pattern of mutation with the cell lines that were impacted by APC knockdown, we did not include the information. I've now looked in the COSMIC database at the recommended genes in the cell lines that are influenced phenotypically by the APC knockdown (BxPC-3, Hs 766T, L3.6pl). The Hs 766T and L3.6pl cells both have mutant KRAS and WT p53, while the Bx-PC3 cells have WT KRAS and mutant p53. Of the three cell lines, only the Hs 766T cells have mutant CDKN2A. In addition, given that the Bx-PC3 cells have WT KRAS, it is unlikely that mutant KRAS is the reason for the independence from the Wnt pathway. However, this would be a good topic on which to follow up. This suggests that further investigation into the cell lines is required to uncover the mechanism by which APC status controls proliferation and migration in pancreatic cancer cells. 

"Beside, in the methods, the source of the utilized cell lines should be stated."

I apologize for omission of this information. We received the cells from Dr. Reginald Hill. This information is now present in the methods and the acknowledgements. 

This manuscript is a resubmission of an earlier submission. The following is a list of the peer review reports and author responses from that submission.

Round 1

Reviewer 1 Report

In the present study, the authors have studied whether APC loss plays a role in pancreatic ductal adenocarcinoma.  It was found that APC knockdown cell lines showed the increased cellular proliferation and no effect on gemcitabine treatment, supportive intervention for cancreatic cancer. However, overall there is something missing throughout the manuscript regarding the overall mechanism being proposed or someho, I am not connecting the dots between APC loss and functional implications regarding pancreatic cancer. Authors simply used pancreatic cell lines to demonstrate their conclusions. At least primary cells or in vivo study such as pancreatic cancer animal model should be performed to prove their hypothesis. How is the experimental design associated with clinical scenario or could APC loss be used clinically in certain situations? Therefore, the conclusion was not supported by the experimental results (cell line study pretty much based on the in-vitro culture).

Reviewer 2 Report

In the present manuscriptthe authors have explored the functional implications of APC loss in pancreatic cancer cells lines. After APC knockdown using lentiviral mediated MISSION shRNA, they performed several in vitro functional assays like proliferation, migration and response to gemcitabine to evaluate the tumorigenic potential of APC loss in this cancer type. Collectively, the findings of this study are interesting for the research field and the experimental design and methodology are reasonable. However, there are some points that need to be addressed:

-the APC status of the examined cell lines should be determined.

-Introduction: they should cite additional work from recent literature when they describe the possible involvement of APC loss in pancreatic cancer.Some examples are:

-Kuo et al. APC haploinsufficiency coupled with p53 loss sufficiently induces mucinous cystic neoplasms and invasive pancreatic carcinoma in mice. Oncogene. 2016 Apr28;35(17):2223-34.

-Jäkel et al. Genome-wide genetic and epigenetic analyses of pancreatic acinar cell carcinomas reveal aberrations in genome stability. Nat Commun. 2017 Nov6;8(1):1323.

-Zhan et al. Germline Variants and Risk for Pancreatic Cancer: A Systematic Review and Emerging Concepts. Pancreas. 2018 Sep;47(8):924-936.

-Yurgelun et al. Germline cancer susceptibility gene variants, somatic second hits, and survival outcomes in patients with resected pancreatic cancer. Genet Med. 2019 Jan;21(1):213-223.